# Research on Classification Method of Building Function Oriented to Urban Building Stock Management

**Bing Xiao [1], Xuexiu Jia [2], Dong Yang [1,\*], Lingwen Sun [1], Feng Shi [3], Qitong Wang [4] and Yongfei Jia [1]**

1   Institute of Science and Technology for Development of Shandong, Qilu University of Technology (Shandong Academy of Sciences), Jinan 250014, China; 10431200838@stu.qlu.edu.cn (B.X.); sunlw@qlu.edu.cn (L.S.); jiayongfei1983@126.com (Y.J.)
2   Sustainable Process Integration Laboratory-SPIL, NETME Centre, Faculty of Mechanical Engineering, Brno University of Technology-VUT Brno, Technická 2896/2, 616 69 Brno, Czech Republic; jia@fme.vutbr.cz
3   School of Environmental Science and Engineering, Qilu University of Technology (Shandong Academy of Sciences), Jinan 250014, China; shifeng1224cn@126.com
4   The Eastern Route of South to North Water Diversion Shandong Main Line Co., Ltd., Jinan 250014, China; qitong1982@126.com
\*   Correspondence: yangdong@qlu.edu.cn

**Abstract:** With the development of human society, the urban population and the urban building stock have been continuously increasing. Environmental issues such as greenhouse gases emissions, air pollution, and construction waste have gradually emerged. Due to the lack of an urban functional area database, it is very time-consuming to manually identify building functional areas. As a result, most of the current research on urban building functions are estimated at a large regional scale or only detailed calculations of individual buildings. The building functions classification method needs to be further improved. Based on the traditional methods, this paper proposes a building function classification method with higher recognition accuracy and is less time-consuming. The method is then applied to a certain area of Chaoyang District, Beijing, for validation and verification. The results show that the urban building function classification method in this paper has a recognition rate of 96.18%, an overall classification accuracy of 94.37%, and a kappa coefficient of 0.9089. The classification results are in good agreement with the virtual interpretation. In addition, automatic classification of building functions is implemented using ArcPy in ArcGIS, which significantly improves the classification efficiency.

**Keywords:** urban building stock; building function classification; POI data

## 1. Introduction

Urban building stock has been growing rapidly with the continuous urbanisation and construction, but the development of a city is at the cost of severe environmental degradation and huge pressure on the ecological environment [1]. The construction, operation, and maintenance, as well as the dismantling, requires a lot of resources and energy and emit greenhouse gases and construction waste. Urbanisation and industrialisation are the main causes of carbon emissions and have a huge impact on global climate change [2]. In addition, 30–40% of urban waste comes from construction waste, most of which ended up in landfills or incarnation plants and caused pollution to the environment [3]. The urban building stock system is a complex system composed of buildings with different structures and functions. The resources and energy consumption and the related environmental impact of buildings with different functions can be quite different. For example, the annual power consumption per unit area of public buildings is 70 to 300 kW, which is 10 to 20 times that of ordinary residential buildings [4]. The building material stock also varies depending on the different functions of the building [5]. As a consequence, urban building function classification is of great importance to improve the accuracy of resource consumption and

environmental impact assessment for urban buildings. There is a need to establish an improved urban building function classification method for the management of urban building inventory, the implementation of the refined classification, and management of the inventory building, which can be useful for the accurate assessment and targeted reduction of greenhouse gas (GHG) emissions and construction waste, and facilitate the realisation of the sustainable development goals (SDGs) proposed by the United Nations in SDG11: Make cities and human settlements inclusive, safe, resilient, and sustainable [6].

In the past few decades, urban construction in some countries and regions has shifted from incremental expansion to inventory optimisation [7]. The resource consumption [8–10] and environmental pollution [11,12] generated by the construction in the process of urbanisation have received more and more attention. Research studies investigating urban building energy consumption [13], building GHG emissions [14], construction waste [15] and architecture material stock [16], and other related topics have been gradually increasing. The essence of these studies is to promote the management of the urban building stock, thereby reducing the waste of resources and environmental pollution and improving the sustainable development of the city. However, as it has been challenging to obtain information on building functions, most of the current studies are carried out investigating only the core building materials, a single building or buildings of a certain type function. For example, Mra [17] conducted a life cycle assessment (LCA) of a typical office building to assess the specific carbon emissions and operational emissions of different building renovation plans. Yu [18] et al. constructed a residential building energy consumption demand model based on the Back Propagation (BP) neural network model and analysed the characteristics of residential building energy consumption in Chongqing, China. Buildings with the same functions have similar energy consumption, carbon emissions, material intensity, etc. Classification of building functions is helpful to improve the accuracy of research on building energy consumption, carbon emission, and risk assessment. For example, Yang [5] established a fundamental database of urban building material strength for residential, public, and industrial buildings in different periods, considering different types of building functions and construction. The database is constructed based on the samples of 813 buildings in 30 provinces in China from 1949 to 2015. Jing et al. [19] investigated the classification of the building considering the forms and functions of Chinese buildings and provided more detailed reference data for earthquake risk analysis and earthquake insurance research. In the existing studies, the classification of building structure and function improves the accuracy of research on urban building energy consumption greatly, building carbon emissions, building material inventory, and building risk assessment. However, traditional building function acquisition methods, such as surveys, questionnaires, etc., are time-consuming and difficult to quickly obtain the buildings function information in large urban areas.

Although there have been some studies investigating the urban building function classification method [20], only a few of them focus on detailed function classification of individual buildings. The building function classification method in most of the existing studies is mainly based on high-resolution remote sensing data and POI (point of interest) data. For example, Li [21] extracted building density and land use data in the study area based on high-resolution remote sensing data, and the function of the building is classified by the Bayesian network method. The economic cost of obtaining high-resolution remote sensing data is relatively high, and the processing of images is also more complicated. In addition, it is difficult to directly identify the functional category of the building with physical information such as the shape, spectrum, space, and texture of the building extracted from the remote sensing image. In contrast, POI data have been gradually used for building function classification thanks to its advantages of low cost and fast acquisition speed. POI data can effectively depict and simulate buildings to achieve three-dimensional visualization of buildings and urban optimization. This Digital Twin technology [22] provides an effective way to realize digital transformation [23], intelligence (such as smart city) [24], and sustainable urban development [25–27]. For example, Qu et al. [28] used POI data

for the kernel density estimation and obtained more accurate results of building function classification. With the advancement of the "big data" era, the POI database representing urban spatial elements is becoming more and more complete and has become to be widely used in the research of urban function zoning and building function classification. Currently, urban building function classification methods based on POI data mainly include kernel density estimation [29] and frequency density ratio method [30], but limited by the density and quantity of POI data, these traditional methods can only be used for rough building function zoning analysis. It is difficult to classify a single building in detail, or the recognition rate and accuracy of the classification is quite limited by the data. It is also time-consuming to manually distinguish by visual interpretation.

This paper aims to improve the existing POI data-based methods of kernel density estimation and frequency density ratio. A building function classification program has been developed with Python using the ArcPy data processing package in ArcGIS, with the aim to automatically process the research data and achieve Automatic classification of building functions in the study area. The detailed classification of urban building functions enables the construction of the urban building functions database, which would facilitate the research of urban building carbon emissions, material inventory, energy consumption, waste, and risk assessment, etc. This would also help to reduce the environmental impact of urbanisation and promote the sustainable development of the city.

## 2. Study Area and Data Processing

### 2.1. Study Area and Data

The study area in this paper is a sub-region in the Chaoyang District, Beijing, China, as shown in Figure 1.

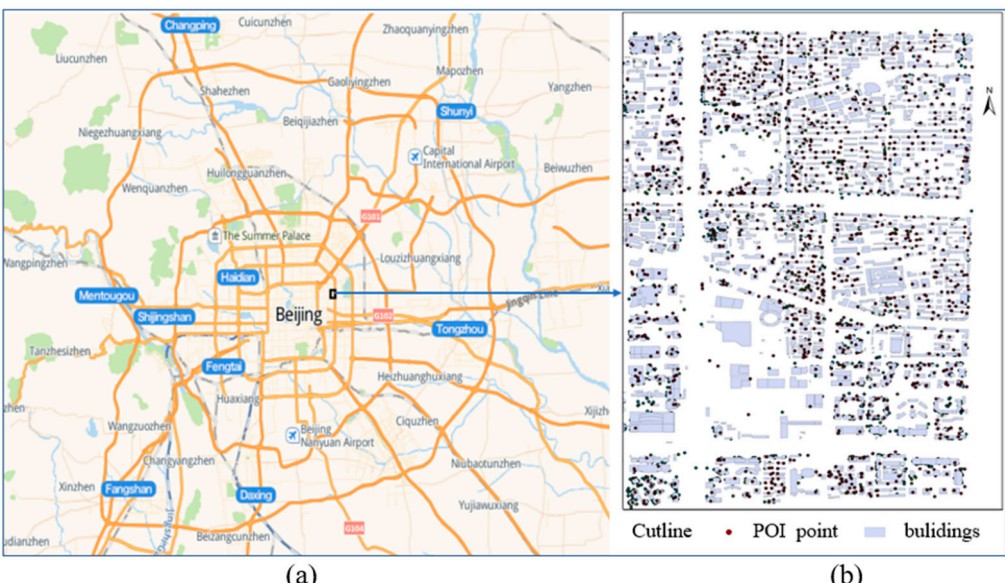

**Figure 1.** (**a**) Selected study area in Chaoyang District, Beijing, China (2021), and (**b**) the POI data map of the selected area.

The dataset used in this study was extracted from the Gaode Map API (https://lbs.amap.com/, accessed on 26 August 2021) in August 2021. The dataset includes 2272 buildings outline data, 65,084 POI data, and 584 real estate data scraped from the real estate information website Anjule (https://beijing.anjuke.com/, accessed on 27 August 2021). The building footprint data cover the building shape and building story information, and the POI data include information such as the name, address, coordinate location, and type. The real estate information includes information such as the name and address, etc., of the building.

*2.2. Data Pre-Processing*

The building footprint data and POI data scraped from the Gaode API need to be pre-processed for the research purpose. The flow chart of the pre-processing is shown in Figure 2.

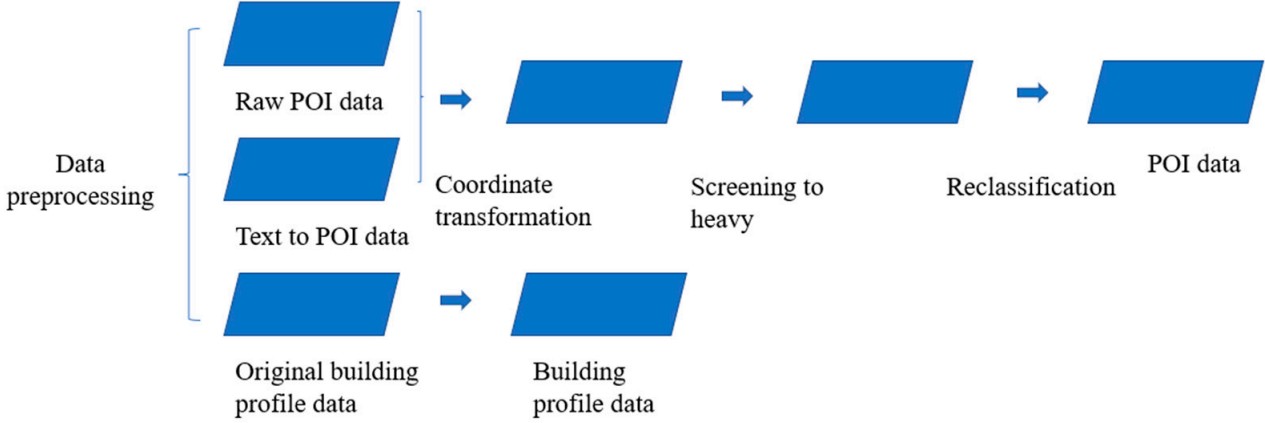

**Figure 2.** Data pre-processing flow chart.

First, as the original data has a position offset, the World Geodetic System—1984 Coordinate System (WGS84) is used for the position correction. At the same time, data deduplication and selection are carried out for the POI data, and the POI data are converted from the real-estate information to obtain the data with the required attributes for this study. Then the POI data are reclassified according to the Chinese National Standard "Current Land Use Classification" (GB/T 21010-2017) [31]. The urban buildings are classified into three groups, as explained in Table 1.

**Table 1.** Building Classification and POI attribute characteristics.

| Building Types | POI Attribute Characteristics |
| --- | --- |
| Residential building | Residential area, district, villa, dormitory, agency dormitory, shanty, etc. |
| Commercial building | Central business district, restaurants, bars, hotels, hotel, shopping mall, shopping malls, furniture, building materials, business hall, hairdressing, beauty, resorts, KTV, Internet cafes, cinemas, concert hall, theatre, massage bath, fitness centre, car sales, car rental, driving, vehicle maintenance and repair, offices, commercial building, architectural decoration, travel agencies, firm, science and technology park, industrial park, etc. |
| Public buildings | Government agencies, organisations, welfare institutions, culture, sports, press and publication, radio and television, exhibition hall, art galleries, general hospital, hospital of a specialised subject hospital, community, animal hospitals, health care and education institutions, institutions of higher education, secondary education, pre-school education, scientific research institutions, adult education, sports, sports venues, tourist attractions, amusement park, parks, museums, galleries, churches, historical sites, science and technology museums, etc. |

Note: For data, please refer to "Classification Standard for Current Land Use (GB/T 21010-2017)".

## 3. Methodology

In this study, the building footprint data and POI data are first scraped from the Gaode Map API, then the real estate information data are obtained from the real estate information network, and the geographic text information is converted into POI data. The variable bandwidth buffer and limited search POI number are used to improve the traditional building function classification method. The frequency density ratio of each type of POI point in the research range of the building is calculated, then the threshold method is used to classify the building function. Finally, the accuracy of the research results with the proposed method is verified. Based on the verified results, the automatic classification of the building function is then designed with Python in ArcPy. The proposed building function classification method is shown in Figure 3.

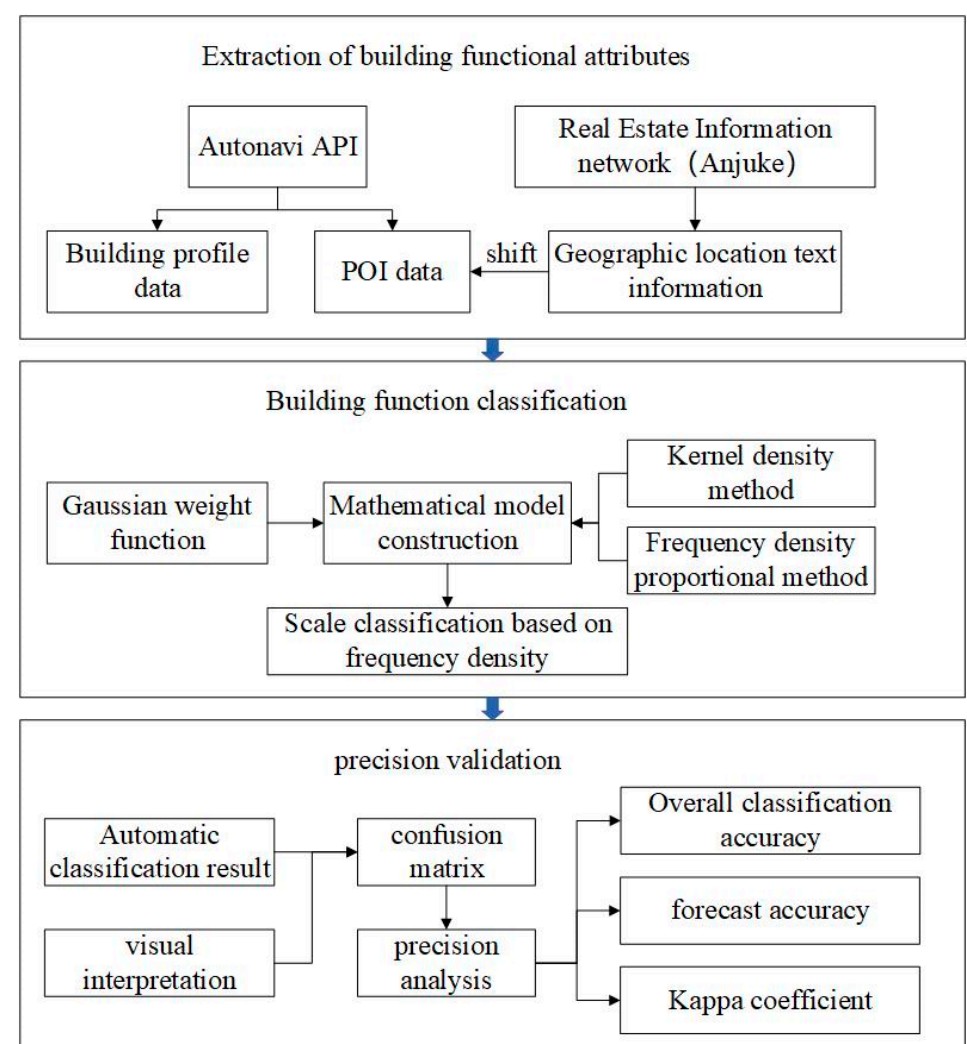

**Figure 3.** Flow chart of building function classification.

### 3.1. Obtaining Building Function Attributes

The buildings functional attributes are mainly obtained from POI data. The POI data scraping was carried out from the Gaode Map API (https://lbs.amap.com/, accessed on 26 August 2021) using a deepweb crawler [32] with the selected keywords according to the national standard "Current Land Use Classification (GB/T 21010-2017)" and the characteristics of building labels (e.g., residential building location information includes the unit, block, floor, building, etc.). Since the marking points of buildings in some areas are not yet complete, there are cases where the POI points of buildings are missing. In

order to increase the number of POI points in the study area and related building function attribute information, real estate information was also crawled from real estate information websites such as Anjuke (https://beijing.anjuke.com/, accessed on 27 August 2021) and other real estate websites. The information such as the name, use, geographic location, etc., of buildings for sale and rent are obtained, and the geographical position text information was converted to POI data using reverse geocoding [33].

### 3.2. Classification of Building Functions

The traditional classification method uses the frequency–density ratio method to identify the function of the building, which requires the POI point representing the building information to fall inside the building footprint. However, in the actual location information labelling and POI data acquisition process, some locations might have a certain offset from the actual location. Meanwhile, the distribution of POI data is uneven. For example, there are more data points in commercial areas and fewer in non-commercial areas. In some cases, there are even no POI points in some remote buildings. As a consequence, using the frequency density ratio method might result in a low recognition rate of building function classification. Kernel density estimation determines the function based on the building function around the studied building. By calculating the kernel density of various types of buildings POI, after many experiments, the kernel density threshold of this type of functional building is obtained for classification after repeated experiments. However, the threshold division usually requires many experiments and also has a greater subjective impact.

Referring to the frequency density ratio method and the kernel density method, the proposed method uses the POI points in the building and the neighbouring POI points to infer the building functional categories in the study area and establish the searching buffer of the neighbouring POI points based on the building outline. The bandwidth of the buffer varies with the density of POI points and the maximum search quantity, and the maximum search distance is set according to the average distance between the geometric centres of the building. That is, the search distance is expanded when the building density in the study area is low and reduced when the building density is high. At the same time, when there is a sufficient number of POI points inside the building to classify the building function, then the searching in the buffer will be stopped. When there are only a few POU points inside the building, the remaining nearest neighbour points are searched from the buffer until there is a sufficient number of points for the classification. The searching will also be stopped when there are not enough POI points both in the building and the largest buffer area. It is critical to assign the weighting factors based on distance, considering that the buildings within shorter distances might have more similar functions. The distance weighting method [34] was applied in this study, and the POI points in the outer contour of the building were given the largest weight with descending weighting factors along the outer contour to the buffer zone. Specifically, the weight of the POI points inside the building is 1 and gradually approaches 0 along the outer contour of the building to the buffer zone. In order to avoid too many POI points being searched and affecting the building classification, in this study, the number of searched POI points is statistically analysed. In other words, the number of the POI points in the building are comprehensively considered, and the appropriate number of neighbouring buildings is selected to assist the classification when there are few POI points in the research building. In this study, the median of the POI points in the research area buildings was the maximum number of selected POI points. The frequency density is calculated as follows:

$$F_i = \frac{\sum_{j=0}^{m} w_j}{N_i} \ (i = 1, 2, 3, \cdots, n; j = 0, 1, 2, \cdots, m) \tag{1}$$

$$w = ae^{-\frac{(x-b)^2}{2c^2}} \tag{2}$$

$$C_i = \& \frac{F_i}{\sum_{i=1}^{n} F_i} \quad (i = 1, 2, 3, \cdots, n) \tag{3}$$

where: $m$ is the number of POI points of type $i$; $w_j$ is the weight function; $Ni$ is the total number of POI points in the $i$-*th* building and in the buffer zone; $F_i$ is the frequency density of type $i$ POI in the building; $x$ is the distance from the POI point to the outer contour of the building; $a$, $b$, $c$ are the constants to adjust the weight; $C_i$ is the proportion of the frequency density of the $i$-*th* type POI in the research building.

The calculation and classification of frequency density of different types of POI were based on the threshold classification method used in the existing studies [35]. Setting a frequency density ratio of 0.5 as the threshold, a specific building function was classified when the ratio was greater than 0.5, and if the building was classified as a Mixed type, then the frequency density ratio was smaller than 0.5. When there was no POI point in the building and the buffer zone, it means that the building function is not able to be classified.

### 3.3. Accuracy Verification

The automatic classification results of this study were compared with the visual interpretation results to verify their accuracy. Three indicators, i.e., the recognition rate, the overall classification accuracy, and the prediction accuracy, were used to verify the classification results. Among these, the classification accuracy and the prediction accuracy were determined using the confusion matrix of the automatic classification results and the visual interpretation results. The indices are calculated as follows.

$$\text{Recognition rate} = \text{number of recognition/total number to be recognised} \tag{4}$$

$$\text{Overall classification accuracy} = \frac{TP + TN}{TP + TN + FP + FN} \tag{5}$$

$$\text{Prediction accuracy rate} = \frac{TP}{TP + FP} \tag{6}$$

where, $TP$ means the true are predicted as true; $FP$ means the true predicted value is false; $FN$ is the false predicted as true; $TN$ is the false predicted as false.

### 3.4. Implementation of Automatic Classification

In order to achieve the rapid classification of the building functions of the study area, this study uses ArcGIS as the platform to compile the processing program of the building function classification on the basis of the improved building function classification method. The processing program is written with Python language with an ArcPy data processing packet in ArcGIS. ArcGIS users can quickly create simple or complex workflows using Python with the help of ArcPy, and develop utilities that can process geological data [36]. The technical process of automatic classification of building functions is shown in Figure 4.

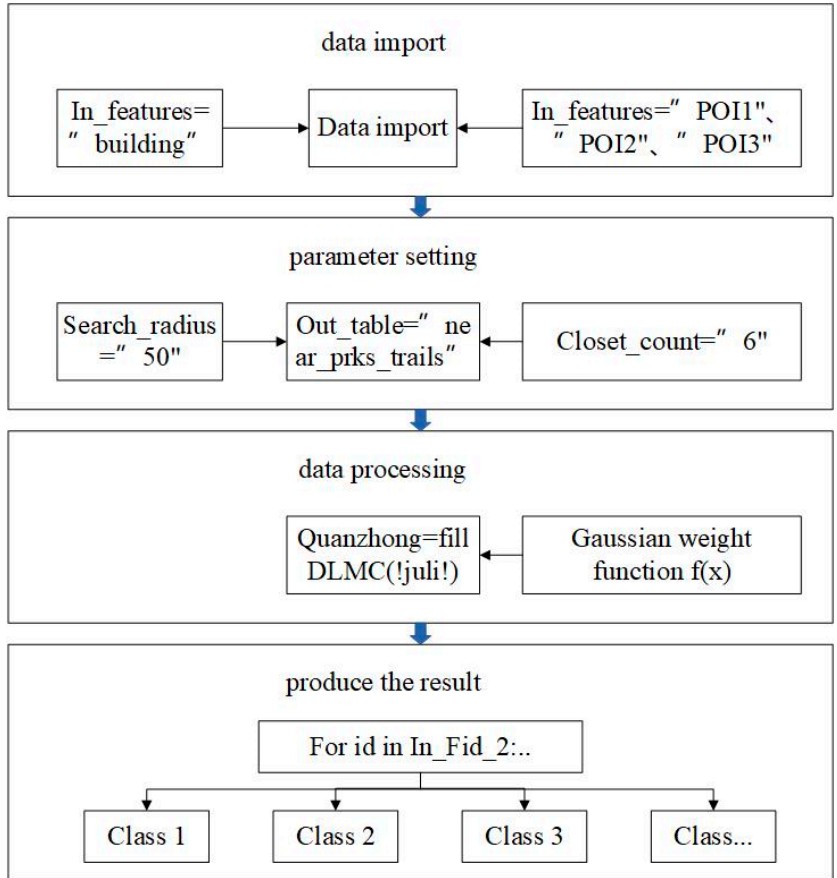

**Figure 4.** Flow chart of the implementation of building function automatic classification.

## 4. Results and Analysis

This paper classifies the buildings in the study area using the improved classification method. The ArcPy automatic building classification tool is established in ArcGIS software and used for the SetNames of the building outline data and POI data table. The weight function of the classification method and the parameters of the farthest search buffer, and the maximum searched POI points are also adjusted. In order to reflect the influence of distance on the weight function, the Gaussian weight function with the curve of fastest descent was adopted, where the weight function parameters were set as a = 1, b = 0, c = 20. The maximum buffer bandwidth was set as 50 m, and the maximum number of search POIs was set to 6. The buildings were divided into residential buildings, commercial buildings, public buildings, and compound buildings according to their functions. Compound buildings refer to a cluster of buildings with different functions, including the following four subgroups: commercial + public buildings (1 + 2), commercial + residential buildings (1 + 3), public + residential buildings (2 + 3), and residential + public + commercial buildings (1 + 2 + 3). The seven categories of building functions are shown in Figure 5.

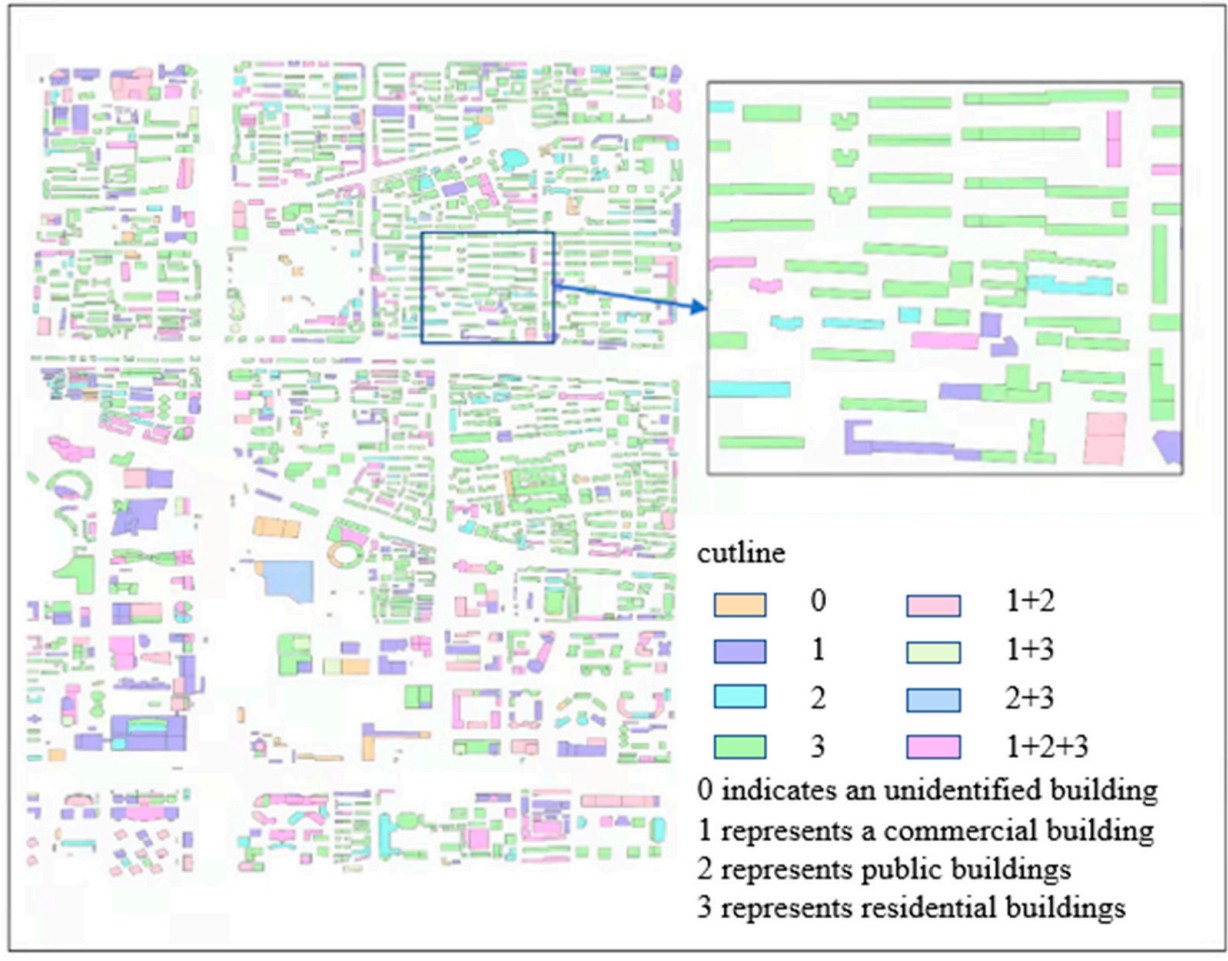

**Figure 5.** Building function classification results.

Half of the automatically classified buildings in the experimental area were randomly selected for comparison with the results of visual classification. In this paper, a total of 969 buildings were selected for the comparison analysis, as shown in Table 2. Results of the confusion matrix showed that the recognition rate of the classification was 96.18%, and the overall classification accuracy is 94.37% with the kappa coefficient of 0.9089, indicating that the automatic classification results are in good agreement with the actual classification. The classification accuracy rate was 95.91% for residential buildings, 95.86% for commercial buildings, 92.13% for public buildings, 90.24% for compound buildings, and 84.85% for unrecognised buildings.

**Table 2.** Confusion matrix table of automatic classification results and visual interpretation results.

| Automatic Classification Results/Block | Visual Interpretation Result/Block | | | | |
|---|---|---|---|---|---|
| | Residential Buildings | Public Buildings | Commercial Buildings | Compound Buildings | Unidentified Buildings |
| Residential buildings | 516 | 3 | 6 | 11 | 2 |
| Public buildings | 3 | 82 | 1 | 2 | 2 |
| Commercial buildings | 2 | 1 | 139 | 2 | 1 |
| Compound buildings | 8 | 3 | 3 | 148 | 2 |
| Unidentified buildings | 3 | 3 | 2 | 1 | 28 |

As the classification method of this article is mainly modified based on the traditional kernel density method and frequency density ratio method, these two methods are also

used for the building function classification, and their results are also compared with the results of the modified method and visual interpretation. The comparison is shown in Table 3.

**Table 3.** Comparison of the results of traditional classification methods, the proposed method in this study, the visual interpretation.

| Methods | Recognition Rate [%] | Overall Classification Accuracy [%] | Time [min] |
|---|---|---|---|
| Frequency density ratio | 50.3 | 43.45 | 13.11 |
| Kernel density | 100 | 52.94 | 47.10 |
| Visual interpretation | 100 | 100 | 1055 |
| The method proposed in this study | 96.18 | 94.37 | 4.37 |

(Note: The recognition rate of visual interpretation was the highest, the recognition rate and overall classification accuracy were theoretically set as 100%).

The recognition rate of the improved method in this study increased 45.82% compared with the traditional frequency density ratio method. This indicated that by setting a variable bandwidth buffer and adding geographic location text information, and converting to POI points, the situation of missing POI points in most areas could be compensated. The proposed method in this study also significantly improved the classification accuracy of classification, with an overall accuracy of 50.92% higher than the frequency density ratio method and 41.43% higher than the kernel density method. In terms of classification efficiency, as shown in Table 3, the classification using ArcPy automatic classification design took about 4.37 min, which was 7.74 min shorter than the frequency density ratio method and 42.73 min shorter than the Kernel density method. Compared with the virtual interpretation, the automatic classification method greatly improved the classification efficiency by just taking 0.4% of the time and ensuring a certain classification accuracy.

## 5. Discussions

The detailed classification of building functions is helpful for the study of urban building energy consumption, carbon emissions, waste generation, and building material inventory, etc. Furthermore, it can also promote the management and planning of urban building inventory and reduce resource waste and environmental pollution. Limited by data and method, the existing studies have not been able to provide accurate and quick building function classification results. The rapid development of information technology enables the detailed classification of building functions. Under the existing technical conditions, data integrity is the main factor limiting the construction of urban building function databases.

This study fills the research gap by the supplement of research data and the improvement of classification methods. The proposed building function classification method of building function classification in this paper has two major advantages compared with the existing methods. First is the improvement of research data integrity and quality. The POI data and building footprint data used in this study are easy-to-obtain, and the real estate information data scraped from the internet is a good supplement to the POI data. In addition, it is relatively easy and time and cost-efficient to scrap urban-level data from the internet compared with remote sensing data. High-resolution remote sensing images can only provide the outline/shape of the building and usually have a higher cost for the higher resolution data. What is more, urban level data require an image mosaic with many remote sensing images to obtain urban level data, as the range of a single remote sensing image is rather small. The process of extracting the outline of the building is also relatively complicated, and finally still needs visual interpretation to classify the urban building functions, which can be time- and energy-consuming for researchers. Second, the recognition rate, total classification accuracy and average prediction accuracy rate experimental results of the proposed classification method are all over 90%, and other main

indicators could also reach a higher level. In addition, the automatic classification program takes only about 4 min to classify the functions of more than 2000 buildings, which implies a remarkable improvement in efficiency compared with the traditional method. With the proposed method, it is possible to quickly and accurately obtain the urban-scale urban function data in the research area, which can be used in the studies of building energy consumption, GHG emissions, building stock analysis, etc., and promote the sustainable development of a city.

As an exploration, this study proposes a method to quickly identify building functions based on POI data. However, due to the limitations of data and methods, as well as the impact of topography and urban layout, some limitations should be noted. It has been found in the experiments that the building outline data and building function attribute data have a considerable influence on the building function classification, which also affects the classification accuracy of building function classification. The basic data affect the building functions classification mainly from the following three aspects. First, there are less or even no POI data of the buildings in remote locations to represent the functional attributes. Although it can be improved by constructing a buffer to search for more POI data, in most such cases, complex types of buildings would be identified, and the classification accuracy is usually low. Second, the current building profile data are only relatively detailed for provincial capitals and some large cities but not sufficiently enough for buildings in small cities. There are still certain limitations for large-scale research at the provincial and national levels. Finally, it was found that the density of POI data can affect the classification results. In this study, the classification algorithm was based on the threshold value, and it was found in the experiment that when the POI data in some buildings and the buffer zone are more concentrated, the POI density ratio values representing different building functions within do not show a significant difference, which can affect the functional building classification. To tackle this issue, only an imitated number of POI data in the research area were selected for the classification. However, it was still found during the verification that there are a few wrong classifications in some cases.

Future research could be carried out in the following aspects: (i) The insufficient number of building footprint data and POI data in the research area can lead to poor classification results. In future studies, additional data such as remote sensing data and street view data will be added to improve the accuracy of district building classification. (ii) In this study, the buildings are only four types of buildings classified into four categories, i.e., residential buildings, public buildings, commercial buildings, and mixed buildings. In future studies, the classification categories can be further subdivided, for example, using building height as a variable. (iii) Apply the improved building function classification method for urban building GHG emissions assessment and refined building management to promote the urban sustainable and healthy development of the city. (iv) In this study, real-time internet data, which can be easily updated, are used for the building function classification. Dynamic databases of for example building functions, GHG emissions, waste generation, and building safety prevention and control can be constructed in future research based on internet data. This can also provide more timely and accurate support for the refined management of urban building stock, etc.

## 6. Conclusions

This research uses data scraping to obtain POI data, building footprint data, and real estate information, improving the existing building function classification method. In addition, the study also compiles the automatic building function classification program based on the ArcGIS platform and implements the automatic classification of the building function in the study area. Comparing the results of automatic classification with the results of visual interpretation, the recognition rate, total classification accuracy, and average prediction accuracy of the buildings in the study area are all above 90%, and the kappa coefficient is 0.9089. The experiment shows that the automatic classification results have good consistency with the actual situation. Compared with other building function

classification methods, the improved method in this paper can quickly and accurately obtain the function information of the buildings in the study area.

As complex open systems, urban cities have many problems and contradictions in their development process. In the era of information, the use of internet data should be strengthened to construct detailed urban building function databases, which can be applied in the research on building carbon emissions, energy consumption, waste management, material inventory, and building risk assessment. These research outcomes could facilitate the urban building stock management and further improve the living standards of residents as well as the sustainable development of a city.

**Author Contributions:** Conceptualization, D.Y. and B.X.; methodology, B.X. and X.J.; software, B.X.; validation, B.X. and X.J.; formal analysis, Q.W.; investigation, L.S. and Y.J.; resources, B.X.; data curation, B.X.; writing—original draft preparation, B.X. and X.J.; writing—review and editing, B.X. and X.J.; visualization, B.X.; supervision, F.S.; project administration, D.Y.; funding acquisition, D.Y. All authors have read and agreed to the published version of the manuscript.

**Funding:** This article is supported by the National Natural Science Foundation of China (Grant No.41971255), The Taishan Scholar Program of Shandong province; The Outstanding Young Fund of Shandong Academy of Science, The Think Tank Project of Shandong Academy of Sciences, Youth Innovation Team of Social Science of Qilu University of Technology. This research was also supported by a project, "Sustainable Process Integration Laboratory-SPIL", project No. CZ.02.1.01/0.0/0.0/15_003/ 0000456 funded by EU as "CZ Operational Programme Research, Development and Education," Priority 1: Strengthening capacity for quality research.

**Data Availability Statement:** Publicly archived datasets: Building profile data, POI data and property information, https://lbs.amap.com/, https://beijing.anjuke.com, accessed on 27August 2021.

**Conflicts of Interest:** The authors declare no conflict of interest.

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
