# Peer review of "Research on Classification Method of Building Function Oriented to Urban Building Stock Management"

_sustainability, doi:10.3390/su14105871_

Round 1

Reviewer 1 Report

The paper is well structured and relevant in the automatisation and digitalization of the urban features recognition field. Its position can be recognized in the number of studies aiming to support the improvement of methods able to recognize building functions. It is in fact focused on developing a new method of automatic recognition of building functions in a specific case study.

The contribution is well designed, and structured and the arguments are well presented and clearly expressed. Tables and figures are also useful for deep comprehension of the authors' position and findings.

I think that the contribution is almost ready for publication. The only suggestion I have is to eventually explore how and if this method can be exported also outside the Chinese context or if it is strictly linked with this country. Additionally, references to the concept of “digital twin” can be included in the literature review to improve the state of the art of the debate on these types of activities.

Reviewer 2 Report

Dear Editor,

Thank you very much for the opportunity to read "Research on Classification Method of Building Function Oriented to Urban Building Stock Management".

The topic of the research is very important.

After reading the paper, I agree with the authors that ”This study fills the research gap by the supplement of research data and the improvement of classification methods. The proposed building function classification method of building function classification in this paper has two major advantages compared with the existing methods. First is the improvement of research data integrity and quality. The POI data and building footprint data used in this study are easy-to-obtain, and the real estate information data scraped from the internet is a good supplement to the POI data. In addition, it is relatively easy and time and cost-efficient to scrap urban-level data from the internet compared with remote sensing data.”

The paper is properly structured and contains sufficient scientific elements with an acceptable novelty.

In particular, to specify which is the part through which the paper brings superior elements in relation to other researchers.

Eventually, the limits of this research should be highlighted.

The bibliography should be extended with some papers (9-10 titles) published in prestigious WoS indexed Journals, in 2021, 2022.

Professor PhD _________________

______________________________________
